# Effect of Gardenia Pomace Supplementation on Growth Performance, Blood Metabolites, Immune and Antioxidant Indices, and Meat Quality in Xiangcun Pigs

**DOI:** 10.3390/ani12172280

**Published:** 2022-09-02

**Authors:** Sen Zou, Changchao Sun, Feng Li, Yingjie Xie, Tong Liang, Yuqing Yang, Baoming Shi, Qingquan Ma, Zhuo Shi, Sa Chai, Anshan Shan

**Affiliations:** College of Animal Science and Technology, Northeast Agricultural University, Harbin 150030, China

**Keywords:** gardenia pomace, pig, growth performance, antioxidant status, immunity, meat quality

## Abstract

**Simple Summary:**

The present study investigated the effects of gardenia pomace on growth performance, blood metabolites, immune and antioxidant indices, and meat quality in Xiangcun pigs. Gardenia pomace supplementation improved the immunity and antioxidant capacities of the Xiangcun pigs and did not negatively affect the growth performance or meat quality of Xiangcun pigs. Gardenia pomace could potentially be used as an unconventional feedstuff in the diet of finishing pigs.

**Abstract:**

To investigate the effect of gardenia pomace (GP) as an unconventional feed of antioxidants, 180 Xiangcun pigs were randomly divided into 3 groups during the finishing period, with 6 replicates per group and 10 pigs per replicate. During the 47-day feeding period, the pigs were fed either a control diet based on corn and soybean meal (control group), or the control diet added with 50 g/kg or 100 g/kg GP (groups GP5 and GP10, respectively). Feed and water were provided ad libitum. One pig per replicate was slaughtered and sampled. The effects on growth performance, meat quality, digestibility, metabolism, and immunity and antioxidant properties of the pigs were investigated. The results showed that GP had no significant effect on the growth performance of Xiangcun pigs. Compared with the control group, the digestibility of crude ash, phosphorus, and crude fibre of pigs in the GP groups improved (*p* < 0.01), and the content of inosinic acid in the longissimus dorsi muscle increased (*p* < 0.05). The addition of GP to the diet significantly increased superoxide dismutase (SOD) levels in the liver and spleen, and glutathione peroxidase (GSH-Px) activity in the longissimus dorsi muscle and spleen (*p* < 0.05). Additionally, it significantly reduced the contents of malondialdehyde (MDA) in the liver and spleen (*p* < 0.05). The GP5 group had a higher inosinic acid content in the longissimus dorsi and lower levels of the inflammatory factor interleukin-2 and interleukin-8 than those in the other groups (*p* < 0.05). The GP10 group had a higher IgA level (*p* < 0.05). Adding different proportions of GP to the diet improved the a* and b* of the longissimus dorsi muscles of Xiangcun pigs (*p* < 0.05). In summary, GP, as an unconventional feed, improved the apparent digestibility of the diet and body antioxidant capacity in Xiangcun pigs during the finishing period and did not negatively affect the growth performance or meat quality.

## 1. Introduction

The search for an alternative feed is of great significance for the sustainable development of animal husbandry [1,2,3]. Some unconventional feeds have essential functions, such as antibacterial and antioxidant activities [4]. Gardenia, a plant of the Rubiaceae family, is rich in iridoid glycosides, chlorogenic acid, crocin, triterpenoids, and has anti-inflammatory, antioxidant, and fibrinolytic activities [5,6,7,8]. Gardenia is mainly distributed in Hunan, Anhui, and other regions in China [9]. Gardenia is the first batch of medicinal and edible plant resources announced by the Ministry of Health in China. Gardenia extract has antioxidant, antibacterial [10], and immune-enhancing functions [11,12]. Ripe gardenia fruit, as a traditional medicinal material in China, protects the liver, promotes bile secretion, and benefits the cardiovascular system. Gardenia pomace (GP) is a by-product produced by industrially extracting biologically active substances such as gardenia yellow pigment, geniposide, and crocin from ripe gardenia fruit through the processes of crushing, extracting, concentrating, and drying [13]. After the above treatments, the GP is dried to less than 10% moisture, which enables it to be stored for a long time at room temperature. There is still small amount of active substances in GP, such as protein and other nutrients. At present, there are few studies on the rational utilisation of GP. Reasonable use of GP is beneficial in many aspects, such as its low carbon footprint which therefore aids in environmental protection. GP is rich in gardenia yellow pigment, in which the major functional components are crocetin and crocetin derivatives [13]. In addition, geniposide in GP has an anti-fibrotic effect on skeletal muscle through suppression of the TGF-β/Smad4 signalling pathway [14]. These findings suggest that GP has the potential to replace part of conventional feed in animal diets and may improve the health status of animals due to its active substances. Based on this hypothesis, this study was conducted to investigate its effects on the growth performance, antioxidant and immunity properties, and meat quality in Xiangcun pigs during the finishing period.

## 2. Materials and Methods

### 2.1. Nutritional Components and Antioxidant Capacity of GP

The GP used in this research was provided by Hunan Hi-Tech Bio-Agro Co., Ltd. (Yueyang, Hunan, China). Crude protein and crude fat contents in GP were determined using the Kjeldahl method [15] and the ether extraction method [16], respectively. The crude fat was extracted with anhydrous ether using a Soxhlet extractor (FT640, Gdanner Instruments Co., Ltd., Guangzhou, China) [17]. The energy content was determined using an oxygen bomb calorimeter (Parr 6300, Parr Instruments, Moline, IL, USA), and the related procedures were described by Shen et al. [18]. Ca and P levels were determined using the potassium permanganate method [19] and colorimetric method, respectively [20]. Extraction and measurement of polysaccharides was performed according to Wang et al. [21]. Extraction and measurement of carotenoids was performed according to Zheng et al. [22]. Extraction and measurement of flavonoids was performed according to Zhang et al. [23]. Extraction and measurement of iridoids was performed according to Zhang et al. [24].

Approximately 1 g of GP powder was added to 40 mL of methanol, weighed, and then sonicated for 40 min. Methanol was added to make up the weight. After mixing, vacuum filtration was performed to obtain a clear filtrate (MEGP) for use. Then, 0.364 g anhydrous sodium acetate was added with 3.2 mL glacial acetic acid at a constant volume of 200 mL, and the pH was adjusted to 3.6 to obtain reagent 1 with 1 mol/L hydrochloric acid solution. Following these steps, 0.078 g dithiothreitol was diluted to 25 mL with 40 mmol/L hydrochloric acid solution to obtain reagent 2; 2.78 g ferric chloride was diluted with ultra-pure water to 50 mL to obtain reagent 3. Reagents 1, 2, and 3 were mixed in a ratio of 10:1:1 to prepare FRAP solution, and 100 μL of MEGP solution was mixed with 3 mL of FRAP solution. The absorbance at 593 nm was measured using a UV spectrophotometer after standing for 20 min. A standard curve was drawn with ferrous sulphate as the standard material, and the antioxidant capacity of the sample was expressed by FRAP value (1 FRAP unit = 1 mmol/L ferrous sulphate). Total antioxidant capacity of MEGP was calculated by the standard curve. Then, 2 mL of MEGP solution was added with 2 mL of ferric sulphite solution with a concentration of 6 mmol/L and 2 mL of hydrogen peroxide solution with a concentration of 6 mmol/L. After reacting in the dark for 10 min, the mixed solution was added to 2 mL of salicylic acid solution with a concentration of 6 mmol/L. After reacting for 30 min in the dark, the absorbance of the mixed solution (X1) was measured at 510 nm using a UV spectrophotometer. The absorbance X2 was measured with double-distilled water instead of MEGP solution, and the absorbance X3 was measured with double-distilled water instead of salicylic acid solution. The formula for calculating hydroxide free radical (OH) scavenging ability is as follows:OH (%) = (X2 − X1 + X3) × 100%/X2(1)

An amount of 2 mL of MEGP solution was added with 2 mL of DPPH–methanol solution with a concentration of 0.2 mmol/L, mixed, and allowed to stand in the dark for 30 min. Absorbance values of the methanol solution (A1) and the reaction solution (A2) were measured at 517 nm using a UV spectrophotometer, respectively. The formula for calculating 1,1-Diphenyl-2-picrylhydrazyl free radical (DPPH) scavenging ability is as follows:DPPH (%) = (A1 − A2) × 100%/A1.(2)

### 2.2. Animals, Experimental Design, and Diets

The Xiangcun pig, which was awarded the national new breed certificate in 2012, is a lean pig bred by cross-breeding with the Taoyuan pig (a local breed in southern China) as the female parent and Duroc pig as the male parent [25]. Xiangcun pigs are basically the same in body shape and appearance and have strong adaptability and resistance [26]. One hundred and eighty castrated male pigs (58.4 ± 4.3 kg, 130 ± 5 days) were selected and randomly divided into three groups with six replicates each, ten pigs per replicate. The control group (CON) was fed a corn–soybean meal basal diet. Other groups (GP5 and GP10) were fed a corn–soybean meal diet containing 50 g/kg or 100 g/kg GP. All diets were formulated in accordance with the recommended amount for the 50–90 kg stage on the China Feeding Standard of Swine (NY/T 65-2004) [27]. Dietary composition and nutritional levels are shown in Table 1. During the 47-day feeding period, food and water were provided ad libitum, but feed was no longer available at 18:00 on day 47 of the feeding experiment. The experiment was conducted at Loudi Base of Xiangcun Pig Breeding (Xiangcun Hi-Tech Co., Ltd., Shaoyang, China).

### 2.3. Growth Performance

The weights of the pigs were recorded at the beginning and end of the feeding experiment, and the daily feeding and remaining amounts were accurately weighed. Average daily gain (ADG), average daily feed intake (ADFI), and feed conversion rate (FCR = g feed/g gain) were calculated.

### 2.4. Apparent Digestibility

Faecal samples for each replicate on days 45, 46, and 47 were collected and mixed. Five percent of the faecal samples were spiked with 10% sulfuric acid solution by weight and then dried for standby. The nutrients in the diet and faeces were evaluated using universal procedures. Crude protein and crude fat contents were determined using the Kjeldahl method [15] and the ether extraction method [16], respectively. The feed samples and faecal samples were dried in an electric blast drying oven at 110 °C for 1–2 h and then pulverised with an ultra-high-speed pulveriser. Approximately 5 g of the sample was digested in a dry digestion tube, and the crude protein content was determined according to the operation method of the Kjeldahl nitrogen analyser (Foss2300, Foss, Hilleroed, Denmark). Approximately 5 g of the sample was collected using a filter paper bucket and the crude fat was extracted with anhydrous ether using a Soxhlet extractor (FT640, Gdanner Instruments Co., Ltd., Guangzhou, China). Approximately 0.5 g of crushed sample was taken and the energy content was determined using an oxygen bomb calorimeter (Parr 6300, Parr Instruments, IL, USA), and the related procedures were described by Shen, Zhu, Liu, Zhang, and Tan [18]. Ca and P levels were determined using the potassium permanganate method [19] and colorimetric method, respectively [20]. Feed digestibility was determined using the acid-insoluble ash method. Acid-insoluble ash was assessed according to Guevara et al. [28]. The acid-insoluble ash content was determined by taking 5 g of the sample in a crucible, placing the sample in a high-temperature electric furnace (SX2-12-12G, Olabor Scientific Instrument Co., Ltd., Jinan, China) for 3 h, adding 25 mL of concentrated hydrochloric acid solution (1:9), and cooling and covering it with a watch glass and boiling it. After cooling for 5 min, the filter paper (ash content of filter paper ≤ 0.01%) was rinsed with hot distilled water until neutral. After drying, the samples were placed in a crucible, burned in a high-temperature electric furnace for 1 h, and weighed after cooling. Feed nutrient digestibility was calculated using the following equation:Feed nutrient digestibility (%) = [1 − (AA_d_/AA_f_) × (NC_f_/NC_d_)] × 100%(3)
where AA_d_ and AA_f_ are acid-insoluble ash content in diet and faeces, respectively. NC_f_ and NC_d_ are nutrient content in faeces and diet, respectively. The recovery rate of acid-insoluble ash in faeces was assumed to be complete.

### 2.5. Blood Biochemistry Indices—Antioxidant and Immunity Indices

On day 46 of the feeding experiment, one pig was randomly selected from each replicate and 10 mL of blood was collected from the jugular vein with sodium heparin vacutainers. After they were kept standing for 30 min, the blood samples were centrifuged at 3500 r/min for 10 min (3 k 30, Sigma, Darmstadt, Germany), and the separated plasma was stored at −20 °C. The levels of total cholesterol (TC), triglycerides (TG), low-density lipoprotein cholesterol (LDL-C), high-density lipoprotein cholesterol (HDL-C), alanine aminotransferase (ALT), alkaline phosphatase (ALP), aspartate transaminase (AST), urea nitrogen (BUN), total protein (TP), albumin (ALB), and uric acid (UA) in plasma samples were determined using commercial test kits (Jiancheng Bioengineering Institute, Nanjing, China) according to the manufacturer’s instructions. The plasma levels of immunoglobulin A (IgA), immunoglobulin M (IgM), immunoglobulin G (IgG), interleukin-2 (IL-2), interleukin-6 (IL-6), interleukin-8 (IL-8), and tumour necrosis factor-α (TNF-α) were determined using ELISA kits (Jiancheng Bioengineering Institute, Nanjing, China), according to the manufacturer’s instructions. The levels of SOD, GSH-Px, and peroxidase (CAT) and total antioxidant capacity (T-AOC) in plasma samples were determined using commercial test kits (Jiancheng Bioengineering Institute, Nanjing, China) according to the manufacturer’s instructions. The minimum detection concentration of IgA, IgM, and IgG kits is less than 0.1 mg/mL. The minimum detection concentration of IL-2, IL-6, IL-8, and TNF-α kits is less than 0.1 pg/mL. The coefficient of variation for MDA, SOD, GSH-Px, CAT, and T-AOC were 1.5%, 1.7%, 3.56%, 1.7%, and 3.6%, respectively.

### 2.6. Slaughtering and Meat Quality

On day 48 of the feeding experiment, one pig was randomly selected from each replicate and stunned by electric shock and slaughtered by bleeding through the cervical vein. The heart, liver, spleen, lungs, and kidneys were removed from one pig of each replicate, and the blood on the surfaces of the organs was wiped with clean absorbent paper. Liver, spleen, and kidney tissues were sampled in cryovials and stored under liquid nitrogen conditions. Approximately 10 g of the left longissimus dorsi muscle between the 10th and 11th ribs, liver, spleen, and kidney were collected. The levels of GSH-Px, MDA, and SOD in tissues were determined using commercial test kits (Jiancheng Bioengineering Institute, Nanjing, China), according to the manufacturer’s instructions.

Approximately 5 g of the left longissimus dorsi muscle were collected for inosinic acid determination by high-performance liquid chromatography, according to Wang et al. [29] (1290 UHPLC, Agilent Technologies Inc., CA, USA). The following formulae were used for the calculations:C_i_ = (C_s_ × A_i_ × M)/(A_s_ × m)(4)
where C_s_ is concentration of inosinic acid in the standard working solution (mg/mL). A_i_ is peak area response value corresponding to inosinic acid. A_s_ is peak area response value corresponding to inosinic acid in the standard working solution. M is total volume of the sample extract (mL), and m is sample mass (g).

The right longissimus dorsi muscle between the 10th and 11th ribs was taken from each slaughtered pig (after 45 min). A part of the sample was dissected, and three points were randomly selected to measure meat colour by a flesh colour instrument (CR-400, Konica Minolta, Tokyo, Japan) immediately. The average value was taken. A part of sample was placed in a glass dish, and the electrode of a pH meter (DHS-2F, Mettler Toledo, Zurich, Switzerland) was inserted into the meat sample to measure pH immediately. Each sample was measured three times, and the average value was taken. Two parts of the sample were stored at 4 °C and measured again at 24 h. Lightness (L*), redness (a*), and yellowness (b*) of the longissimus dorsi muscle were measured using the flesh colour instrument (CR-400, Konica Minolta, Tokyo, Japan), as assessed by Babic et al. [30]. The pH value was measured using the pH meter (DHS-2F, Mettler Toledo, Zurich, Switzerland), according to the method of Li et al. [31]. About 40 g of the longissimus dorsi muscle was used to determine drip loss, according to Przybylski et al. [32]. The following formulae were used for the calculations:Drip loss (%) = [(W_b_ − W_a_)/W_b_] × 100%(5)
where W_a_ and W_b_ are weight of meat before and after hanging, respectively. Approximately 40 g of the longissimus dorsi muscle was hung in a bottle and placed at 4 °C in a refrigerator for 24 h. The longissimus dorsi muscle was used for cooking loss determination, according to the method of Bertram et al. [33]. The following formulae were used for the calculations:Cooking loss (%) = [C_b_ − C_a_/C_b_] × 100%(6)
where C_b_ and C_a_ are the weight of the meat before and after cooking, respectively. Approximately 40 g of the longissimus dorsi muscle was sealed and placed in a water bath at 75 °C for 30 min. The meat samples were taken out, cooled under running water at 15 °C for 40 min, and the surface water was absorbed using paper. The shearing force of the meat samples was measured using a universal Warner-Bratzler tester (C-LM3, Northeast Agricultural University, Harbin, China) according to the manufacturer’s instructions [34].

### 2.7. Statistical Analysis

The mean of each replicate was the experimental unit for the growth performance and apparent digestibility. The pig was the experimental unit for other data of the present study. All data were analysed by one-way ANOVA using SPSS software (version 22.0; SPSS Inc., Chicago, IL, USA). Significant differences in means among treatments were tested by Duncan’s multiple range tests. The results are expressed as the means, *p*-value, and pooled SEM (standard error of the mean). Differences were considered statistically significant at *p* < 0.05. Figures were generated by Graph-Pad Prism 8.0 software (GraphPad Software, San Diego, CA, USA).

## 3. Results

### 3.1. Nutritional Components and Antioxidant Capacity of GP

Nutritional components in GP are shown in Table 2. The GP was rich in crude protein, crude fibre, crude fat, total energy, and carotenoids. Polysaccharides, flavonoids and iridoids were still contained in the GP.

Antioxidative capacity of methanol extract from GP is shown in Table 3. The GP crude extract had strong scavenging ability to DPPH and OH radicals, and the scavenging rates of the two radicals were 96.60% and 84.74%, respectively.

### 3.2. Growth Performance

The effects of GP on the growth performance of the Xiangcun pigs are presented in Table 4. Compared with the control group, GP did not significantly affect the final weight, ADG, ADFI, or FCR of the Xiangcun pigs (*p* > 0.05).

### 3.3. Apparent Digestibility

The effect of GP on apparent dietary digestibility is shown in Table 5. Compared with the control group, the digestibility of crude fibre, crude ash, and phosphorus increased significantly in the GP5 and GP10 groups (*p* < 0.01). However, the total energy digestibility of the pigs in the GP10 group was reduced (*p* < 0.05). There was no significant effect of GP on the apparent digestibility of crude protein and calcium in Xiangcun pigs (*p* > 0.05).

### 3.4. Blood Biochemistry Indices—Antioxidant and Immunity Indices

The effects of GP on plasma biochemical indices of the Xiangcun pigs are shown in Table 6. Compared with the control group, the levels of UA, TG, and LDL-C in the plasma samples of the GP5 group were higher (*p* < 0.05). In the GP10 group, only the plasma HDL-C level was reduced (*p* < 0.05).

The effects of adding GP to the diet on immune indices in the plasma of the Xiangcun pigs are shown in Table 7. Compared to the control group, the levels of IgM, IL-2, and IL-8 in the plasma samples of the Xiangcun pigs in the GP5 group were reduced (*p* < 0.05). There was no significant effect observed on the levels of IgM, IL-2, and IL-8 in the plasma samples of GP10 group (*p* > 0.05).

The effects of GP on the antioxidant indices in the plasma samples of the Xiangcun pigs are shown in Table 8. The enzyme activity of GSH-Px in the plasma samples of the GP10 group was increased (*p* < 0.05). However, there was no significant effect on the enzyme activity of GSH-Px in the plasma samples of the GP5 group (*p* > 0.05). There were no significant effects of GP5 and GP10 on CAT and T-AOC in the plasma samples of the Xiangcun pigs (*p* > 0.05). Compared to the GP10 group, the GP5 group showed a higher CAT enzyme activity (*p* < 0.05).

### 3.5. Slaughtering and Meat Quality

The effects of adding GP to the diet on the antioxidant indices of the organs and longissimus dorsi muscle of the Xiangcun pigs are shown in Figure 1. Compared with the control group, the liver and spleen MDA contents in the GP5 and GP10 groups were reduced (*p* < 0.05). The MDA content in the longissimus dorsi muscles of the Xiangcun pigs showed reduction in the GP5 group (*p* < 0.05). The activity of SOD in the liver and spleen of the Xiangcun pigs were increased in the GP groups (*p* < 0.05). The GSH-Px content in the spleens and longissimus dorsi muscles of the GP5 and GP10 groups increased (*p <* 0.05). However, the GSH-Px content in the kidneys of the GP10 group decreased.

The effect of GP on inosinic acid in the longissimus dorsi muscle of Xiangcun pigs is shown in Figure 2. Compared to the control group, the inosinic acid levels in the longissimus dorsi muscles of the Xiangcun pigs in the GP5 and GP10 groups significantly increased (*p* < 0.01).

The effects of GP on the longissimus dorsi muscle quality of Xiangcun pigs are shown in Table 9. Compared to the control group, the a* of the longissimus dorsi muscle of the GP5 group increased after 45 min of slaughter (*p* < 0.05) but decreased in the GP10 group (*p* < 0.05). The a*, b*, and pH values of the longissimus dorsi muscles in the GP5 group increased after 24 h of slaughter (*p* < 0.01). There was no significant effect of GP on the shear force, drip loss, cooking loss, L*, and pH of the longissimus dorsi muscles of Xiangcun pigs 45 min after slaughter (*p* > 0.05).

## 4. Discussion

Gardenia is still cultivated for ornamental use. Moreover, it is used in traditional Chinese medicine [35,36]. The active constituents of gardenia fruits are iridoid glycosides such as geniposide, gardenoside, and gardoside [37]. GP is the residue remaining after extracting the active substances and gardenia oil from the gardenia fruit [38]. Our research showed that GP had a proximate composition with 22.85 MJ/kg gross energy, 111.4 g/kg crude protein, 191.2 g/kg crude fat, 13.2 g/kg calcium, 3.6 g/kg phosphorus, and 276.4 g/kg crude fibre, which was like rice bran. Moreover, carotenoids, polysaccharides, flavonoids, and iridoids were still contained in the GP. According to relevant research, there was a certain antioxidant capacity in gardenia polysaccharides, and the scavenging rates of DPPH, OH, and superoxide anion polysaccharides were 21.650 g/kg, 70.97%, and 19.26%, respectively [21]. The results of this experiment also showed that the crude extract of GP had strong scavenging ability to DPPH and OH free radicals, which further proved the antioxidant capacity of GP. Iridoids have antidepressant and anti-inflammatory activities, and they are beneficial to the nervous, cardiovascular, and digestive systems [39]. Si et al. [40] reported that geniposide, the main component of iridoids, could promote the proliferation and differentiation of neural stem cells in vitro. The intestinal microbial hydrolysate of geniposide, genipin, was believed to cause liver toxicity [41]. From the perspective of production performance, there was no effect at the end of growth on the average daily gain and feed coefficient of the Xiangcun pigs fed 50 g/kg and 100 g/kg diets. Simultaneously, the digestibility of crude ash and phosphorus in the experimental group of the Xiangcun pigs increased. Formic acid contained in GP might promote the secretion of digestive juices, thereby enhancing the digestive absorption capacity of the Xiangcun pigs [42,43]. In addition, the extractive in GP could regulate the balance of animal intestinal flora, resulting in an increase in the digestibility of nutrients in the feed [44]. This might also be because Xiangcun pigs have strong resistance to roughage and can effectively tolerate cellulose in the diet. At a 100 g/kg GP diet, Xiangcun pigs might not be able to rapidly adapt to the high crude fibre content in the diet, resulting in the decrease in total energy digestibility to a certain extent. The growth status of the pigs also showed no significant difference between the experimental and control groups. GP diets had no adverse effects on growth performance and feed digestibility. These results proved that Xiangcun pigs could adapt to GP feeds. There was no toxicity to GP for Xiangcun pigs in this experiment.

Animal blood indicators reflect the health and nutritional metabolism of animals. The experimental results showed that the plasma TG levels were reduced by feeding GP to the Xiangcun pigs. Tang et al. [45] showed that lipoprotein-associated phospholipase A2 in pigs was related to TG levels and could interact with HDL-C and LDL-C. As assessed in this experiment, HDL-C levels were reduced in the 100 g/kg diet and LDL-C levels were increased in the 50 g/kg diet. This might be because certain active ingredients in GP promoted the expression of lipoprotein-associated phospholipase A2. TC is mainly composed of low-density lipoprotein, high-density lipoprotein, and very-low-density lipoprotein [46]. The result of the GP5 group showed that both HDL-C and LDL-C were significantly increased, while the level of TC remained unchanged. This might be related to an increase in HDL-C and LDL-C and a decrease in very-low-density lipoprotein. Excessive accumulation of serum enzymes, such as AST and ALP, usually indicates liver damage [47]. In this research, low-dose GP had no toxic damage to the liver, and there were other active substances (such as geniposide, organic acids, etc.) in the GP that could enhance liver function. Immune indicators in the blood primarily include inflammatory cytokines (interleukins and tumour necrosis factors) and immunoglobulins (IgA, IgM, and IgG) [48]. Shi et al. [49] reported that geniposide could reduce the expression of inflammatory factors (TNF-α and IL-6) by inhibiting the overexpression of inducible nitric oxide synthase and cyclooxygenase-2. In this research, IgA levels were increased in the GP10 group, and IL-2 and IL-8 levels were reduced in the GP5 group. However, the level of IgG was reduced in the GP5 and GP10 groups and the level of IgM was reduced in the GP5 group. This might be due to that the overexpression of inducible nitric oxide synthase and cyclooxygenase-2 was reduced by geniposide, which in turn meant the expression of inflammatory factors was reduced, which led to reductions in IgG and IgM levels [50]. As assessed this experiment, the plasma SOD and GSH-Px levels were increased by feeding a 100 g/kg diet, and the plasma SOD level was also increased in the 50 g/kg diet. Reddy et al. [10] reported that gardenia extract had a strong antioxidant capacity. The above experimental results showed that antioxidant performance could be improved by the active ingredients in GP.

Furthermore, we studied the antioxidant properties of the liver, kidney, spleen, and longissimus dorsi in Xiangcun pigs. MDA is a biomarker of oxidative stress. If its content is higher than that in healthy individuals, it is considered harmful to the body [51]. In the present study, GP supplementation decreased MDA levels in the liver, spleen, and longissimus dorsi muscle. SOD and GSH-Px, which are essential components in the antioxidant system of mammals, were significantly improved in the groups fed with GP. The levels of inosinic acid increased in the GP diet groups. Pu et al. [52] reported that the inosinic acid in the muscle of Nile tilapia could be increased, and its flavour was improved by astragalus, Angelica, and other Chinese herbal medicines. GP used in this research, as a by-product of ripe gardenia fruit, contained similar effective substances, which increased the expression of inosinic acid. These results were in line with that of Xiao et al. [6], who found that antioxidant capacity could be improved by geniposide. At the time of slaughter, the pigs were in good growth condition, and no lesions were found in the liver, kidney, spleen, or longissimus dorsi. Meat colour, including L*, a *, and b *, is a critical evaluation criterion for consumers to purchase high-quality meat. The experimental results showed that a*, b*, and pH increased in the GP5 group. In general, meat with drip loss > 6.0% and pH < 5.9 after 45 min of slaughter is called PSE meat (pale, soft, exudative). Meat with drip loss < 2.0% and pH > 6.5 after 24 h of slaughter is called DFD meat (dark, firm, and dry) [53]. Compared with the control group, although the pH was significantly increased in the GP-supplemented group, it did not reach the pH level of DFD meat. Compared with the control group, the drip loss in the 50 g/kg GP group tended to decrease. Geniposide of GP has antioxidant capacity, which can remove free radicals produced by the body, enhance the activity of antioxidant enzymes in the body, change the permeability of cell membranes, protect the normal function of cell membranes, and reduce drip loss [54]. In this research, GP had no effect on L* and the shearing force in the Xiangcun pigs.

## 5. Conclusions

GP supplementation in the diet of finishing Xiangcun pigs improved the apparent digestibility of crude fibre, crude ash, and phosphorus. The level of a* in meat and the antioxidant capacity of plasma, liver, spleen, and longissimus dorsi muscle were also significantly increased. The levels of IL-2 and IL-8 in the blood were reduced by the GP diet, and the level of IgA was increased. The GP diet did not negatively affect the growth performance or meat quality of Xiangcun pigs.

## Figures and Tables

**Figure 1 animals-12-02280-f001:**
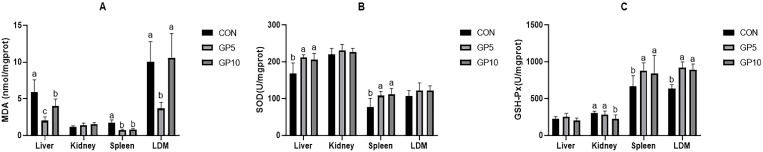
Effects of gardenia pomace on contents of MDA (**A**), SOD (**B**), and GSH-Px (**C**) in the organs and the longissimus dorsi muscle of Xiangcun pigs. CON, basal diet group; GP5, 50 g/kg gardenia pomace treatment group; GP10, 100 g/kg gardenia pomace treatment group; LDM, longissimus dorsi muscle. The pig was the experimental unit, *n* = 6. ^a^, ^b^, ^c^ Values within a row with different superscripts letter differ significantly at *p* < 0.05.

**Figure 2 animals-12-02280-f002:**
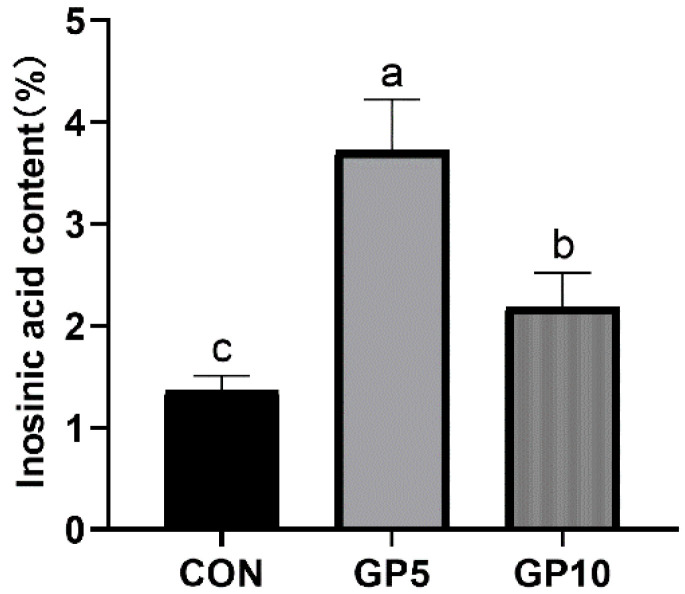
Effect of gardenia pomace on inosinic acid levels in the longissimus dorsi muscle of Xiangcun pigs (freeze-dried basis, mg/g). CON, basal diet group; GP5, 50 g/kg gardenia pomace treatment group; GP10, 100 g/kg gardenia pomace treatment group. The pig was the experimental unit, *n* = 6. ^a^, ^b^, ^c^ Different letters indicate significant differences (*p <* 0.05).

**Table 1 animals-12-02280-t001:** Composition and nutritional level of the diets of Xiangcun pigs.

Items	0–25 Day			26–47 Day		
	CON	GP5	GP10	CON	GP5	GP10
Corn (g/kg)	732.9	716.5	670.4	778.6	761.3	715.3
Soybean meal (g/kg)	198.3	201.5	199.0	154.2	156.9	154.5
Gardenia pomace (g/kg)	-	50.0	100.0	-	50.0	100.0
Wheat bran (g/kg)	35.1	-	-	34.0	-	-
Methionine (g/kg)	-	0.1	0.2	-	0.2	0.3
Lysine (g/kg)	3.3	3.3	3.3	3.6	3.6	3.7
Threonine (g/kg)	0.3	0.2	0.1	0.4	0.3	0.2
Tryptophan (g/kg)	-	-	0.1	-	0.1	-
Limestone (g/kg)	6.5	4.6	3.2	6.4	4.5	3.0
CaHPO_4_ (g/kg)	9.6	9.8	9.7	8.8	9.1	9.0
Sodium chloride (g/kg)	4.0	4.0	4.0	4.0	4.0	4.0
Premix ^1^ (g/kg)	10.0	10.0	10.0	10.0	10.0	10.0
Nutritional level ^2^						
DE (Mcal/kg)	3.3	3.3	3.2	3.3	3.3	3.2
Crude protein (g/kg)	159.8	159.8	159.8	144.5	144.5	144.4
Crude fibre (g/kg)	25.8	35.4	46.5	23.9	33.5	44.6
NDF (g/kg)	109.6	123.2	147.7	107.4	121.3	145.9
ADF (g/kg)	43.0	58.0	75.8	39.9	54.9	72.7
Lys (g/kg)	9.8	9.8	9.8	9.0	9.0	9.0
Met (g/kg)	2.6	2.6	2.7	2.4	2.5	2.6
Met + Cys (g/kg)	5.4	5.4	5.4	5.1	5.1	5.1
Thr (g/kg)	6.0	6.0	6.0	5.4	5.4	5.4
Try (g/kg)	1.6	1.6	1.6	1.4	1.4	1.3
Calcium (g/kg)	6.0	6.0	6.0	5.6	5.6	5.6
Total phosphorus (g/kg)	5.7	5.6	5.6	5.4	5.3	5.3
Available phosphorus (g/kg)	3.0	3.0	3.0	2.8	2.8	2.8
Sodium (g/kg)	1.7	1.7	1.7	1.7	1.7	1.7
Chlorine (g/kg)	2.7	2.7	2.7	2.7	2.7	2.7
Carotene ^3^ (g/kg)	-	3.0	6.1	-	3.0	6.1
Iridoids ^3^ (mg/kg)	-	43.0	86.0	-	43.0	86.0
Flavonoids ^3^ (mg/kg)	-	19.5	39.0	-	19.5	39.0
Polysaccharide ^3^ (g/kg)	-	0.2	0.3	-	0.2	0.3

CON, basal diet group; GP5, 50 g/kg gardenia pomace treatment group; GP10, 100 g/kg gardenia pomace treatment group; DE, digestible energy; NDF, neutral detergent fibre; ADF, acid detergent fibre. ^1^ Premix was provided per kilogram of ration: CuSO_4_·5H_2_O, 20 mg; FeSO_4_·7H_2_O, 90 mg; ZnO, 75 mg; MnSO_4_·H_2_O, 40 mg; KI, 0.2 mg; NaSeO_3_·5H_2_O, 0.4 mg; vitamin A, 10 000 IU; vitamin D3, 2 000 IU; vitamin E, 20 IU; vitamin K3, 1 mg; vitamin B1, 1.8 mg; vitamin B2, 5 mg; vitamin B6, 1.5 mg; vitamin B12, 0.01 mg; niacin, 20 mg; pantothenic acid, 15 mg. ^2^ The data are calculated values according to Feeding Standards of the Ministry of Agriculture of China (NY/T65-2004). ^3^ Active substance content, content in GP.

**Table 2 animals-12-02280-t002:** Nutritional components of gardenia pomace (air-dried basis).

Item	Content
Gross energy (MJ/kg) ^1^	22.85
Crude protein (g/kg) ^1^	111.40
Crude fat (g/kg) ^1^	191.20
Ca (g/kg) ^1^	13.20
P (g/kg) ^1^	3.60
Crude fibre (g/kg) ^1^	276.40
Carotenoids (g/kg) ^2^	60.78
Polysaccharides (g/kg) ^2^	3.33
Flavonoids (g/kg) ^2^	0.39
Iridoids (g/kg) ^2^	0.86

^1^ Each value represents the mean of 3 replicate values. ^2^ Each value represents the mean of 4 replicate values.

**Table 3 animals-12-02280-t003:** Antioxidative capacity of methanol extract from gardenia pomace.

Item	Content ^1^
Total antioxidant capacity (FRAP)	0.16
OH (%)	84.74
DPPH (%)	96.60

1 FRAP unit = 1 mmol/L ferrous sulphate. OH, hydroxide free radical; DPPH, 1,1-Diphenyl-2-picrylhydrazyl free radical. ^1^ Each value represents the mean of 5 replicate values.

**Table 4 animals-12-02280-t004:** Effects of gardenia pomace on growth performance of Xiangcun pigs.

Items	CON	GP5	GP10	SEM	*p*-Value
Initial weight (kg)	58.00	58.93	58.27	0.498	0.757
Final weight (kg)	82.15	82.68	84.00	0.428	0.199
ADG (kg/d)	0.51	0.51	0.55	0.015	0.493
ADFI (kg/d)	1.91	1.87	1.83	0.015	0.072
FCR	3.79	3.73	3.37	0.114	0.285

CON, basal diet group; GP5, 50 g/kg gardenia pomace treatment group; GP10, 100 g/kg gardenia pomace treatment group; ADG, average daily gain; ADFI, average daily feed intake; FCR, feed conversion rate. The mean of each replicate was the experimental unit, *n* = 6.

**Table 5 animals-12-02280-t005:** Effects of gardenia pomace on apparent digestibility in Xiangcun pigs.

Items	CON	GP5	GP10	SEM	*p*-Value
Crude protein (%)	66.20	62.42	62.65	0.812	0.088
Crude fibre (%)	34.61 ^b^	52.97 ^a^	58.23 ^a^	2.879	<0.001
GE (MJ/kg)	75.63 ^a^	73.81 ^a,b^	71.30 ^b^	0.632	0.012
Crude ash (%)	54.47 ^c^	60.63 ^b^	71.70 ^a^	1.946	<0.001
Calcium (%)	12.61	11.86	14.06	1.533	0.865
Phosphorus (%)	12.17 ^c^	25.32 ^b^	36.71 ^a^	2.775	<0.001

CON, basal diet group; GP5, 50 g/kg gardenia pomace treatment group; GP10, 100 g/kg gardenia pomace treatment group. The mean of each replicate was the experimental unit, *n* = 6. ^a^, ^b^, ^c^ Values within a row with different superscript letters differ significantly at *p* < 0.05.

**Table 6 animals-12-02280-t006:** Effect of gardenia pomace on the blood plasma biochemical indices of Xiangcun pigs.

Items	CON	GP5	GP10	SEM	*p*-Value
ALB (g/L)	32.15	26.12	21.71	1.933	0.078
TP (gprot/L)	57.91	59.69	54.45	1.656	0.447
BUN (mmol/L)	6.68	5.26	7.51	0.542	0.239
UA (µmol/L)	93.64 ^b^	219.53 ^a^	148.85 ^a,b^	21.535	0.046
TG (mmol/L)	1.19 ^b^	2.70 ^a^	0.99 b	0.222	<0.001
TC (mmol/L)	2.53	2.21	3.03	0.257	0.440
LDL-C (mmol/L)	0.99 ^b^	2.91 ^a^	1.49 ^b^	0.272	0.004
HDL-C (mmol/L)	2.93 ^a^	3.50 ^a^	0.97 ^b^	0.407	0.023
ALP (U/L)	151.68	81.40	90.71	19.595	0.299
ALT (IU/L)	15.97	8.86	10.63	2.016	0.328
AST (IU/L)	41.80	57.93	57.75	8.869	0.721

CON, basal diet group; GP5, 50 g/kg gardenia pomace treatment group; GP10, 100 g/kg gardenia pomace treatment group; ALB, albumin; TP, total protein; BUN, urea nitrogen; UA, uric acid; TG, triglycerides; TC, total cholesterol; LDL-C, low-density lipoprotein cholesterol; HDL-C, high-density lipoprotein cholesterol; ALP, alkaline phosphatase; ALT, alanine aminotransferase; AST, aspartate transaminase. The pig was the experimental unit, *n* = 6. ^a^, ^b^ Values within a row with different superscript letters differ significantly at *p* < 0.05.

**Table 7 animals-12-02280-t007:** Effect of gardenia pomace on the blood plasma immune indices of Xiangcun pigs.

Items	CON	GP5	GP10	SEM	*p*-Value
IgA (mg/mL)	709 ^b^	672.25 ^b^	867.95 ^a^	1.417	0.044
IgG (mg/mL)	43.52 ^a^	30.52 ^b^	30.47 ^b^	1.726	<0.001
IgM (µg/mL)	32.87 ^a^	24.52 ^b^	28.83 ^a,b^	1.417	0.044
IL-2 (pg/mL)	439.88 ^a^	309.92 ^b^	415.52 ^a^	32.397	0.021
IL-6 (pg/mL)	1104.46	896.89	1122.50	60.699	0.271
IL-8 (pg/mL)	74.03 ^a^	55.13 ^b^	59.74 ^a,b^	2.458	0.001
TNF-α (pg/mL)	311.12	246.76	305.61	44.353	0.728

CON, basal diet group; GP5, 50 g/kg gardenia pomace treatment group; GP10, 100 g/kg gardenia pomace treatment group; IgA, immunoglobulin A; IgG, immunoglobulin G; IgM, immunoglobulin M; IL-2, interleukin-2; IL-6, interleukin-6; IL-8, interleukin-8; TNF-α, tumour necrosis factor-α. The pig was the experimental unit, *n* = 6. ^a^, ^b^ Values within a row with different superscript letters differ significantly at *p* < 0.05.

**Table 8 animals-12-02280-t008:** Effect of gardenia pomace on the blood plasma antioxidant indices of Xiangcun pigs (U/mL).

Items	CON	GP5	GP10	SEM	*p*-Value
GSH-Px	445.80 ^b^	467.86 ^b^	720.43 ^a^	42.157	0.020
CAT	3.44 ^a,b^	4.45 ^a^	2.57 ^b^	0.319	0.045
T-AOC	5.38	7.55	3.62	0.738	0.085
SOD	167.93	303.57	302.94	27.757	0.061

CON, basal diet group; GP5, 50 g/kg gardenia pomace treatment group; GP10, 100 g/kg gardenia pomace treatment group; GSH-Px, glutathione peroxidase; CAT, peroxidase; T-AOC, total antioxidant capacity; SOD, superoxide dismutase. The pig was the experimental unit, *n* = 6. ^a^, ^b^ Values within a row with different superscript letters differ significantly at *p* < 0.05.

**Table 9 animals-12-02280-t009:** Effects of gardenia pomace on the meat quality of Xiangcun pigs.

Items	CON	GP5	GP10	SEM	*p*-Value
45 min					
L*	40.82	41.01	39.80	0.831	0.832
a*	6.82 ^b^	7.83 ^a^	5.66 ^c^	0.267	0.003
b*	5.01	4.60	3.99	0.177	0.064
pH	6.41	6.57	6.49	0.044	0.350
24 h					
L*	46.15	46.86	45.96	0.598	0.817
a*	6.88 ^b^	10.91 ^a^	7.91 ^b^	0.426	<0.001
b*	6.97 ^b^	9.83 ^a^	7.64 ^b^	0.333	<0.001
pH	5.95 ^b^	6.10 ^a^	5.87 ^b^	0.027	0.001
Drip loss (%)	1.26	0.62	0.88	0.116	0.068
Cooking loss (%)	19.91	15.52	18.25	1.305	0.406
Shear force (N)	68.53	74.27	62.21	4.232	0.536

CON, basal diet group; GP5, 50 g/kg gardenia pomace treatment group; GP10, 100 g/kg gardenia pomace treatment group; L*, lightness; a*, redness; b*, yellowness. The pig was the experimental unit, *n* = 6. ^a^, ^b^, ^c^ Values within a row with different superscript letters differ significantly at *p* < 0.05.

## Data Availability

The data presented in this study are available in the present article and is shared with consent and in accordance with all authors.

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
