# Peer review of "Effect of Gardenia Pomace Supplementation on Growth Performance, Blood Metabolites, Immune and Antioxidant Indices, and Meat Quality in Xiangcun Pigs"

_animals, 2022, doi:10.3390/ani12172280_

Round 1
Reviewer 1 Report
animals-1856931
Title: Gardenia pomace as a nontraditional feedstuff: growth performance, meat quality, blood metabolites, immune and antioxidant indices in Xiangcun pigs
This paper studies the effects of Gardenia Pomace as a nontraditional feedstuff on growth performance, meat quality, blood metabolites, Immune and Antioxidant Indices in Xiangcun Pigs. The research content is rich, but the objects are not what most people care about. Moreover, the materials and methods were not fully explained, the source of the test materials and the pig breed was unknown, the information of the ELISA kit used was not provided in detail. The statistical methods were also too simple.
Line 55-62, These unpublished data should be in this article, should not be in the supplementary Table S1. Because these are very important information for this study. Please move it into the Results.
Many readers don't know much about Gardenia, so please introduce this plant in detail. What kind of plant is it? Was the root, stem, leaf, flower, or fruit used in this study? How to harvest processing the Gardenia? Finally, the GP used in the experiment was obtained? I don’t know. These should be explained in detail, otherwise the experiment cannot be repeated by others.
Line 73, Please introduce Xiangcun pigs here. Because most readers don't know this pig breed. Were they purebred Chinese native pig breed? Or a crossbreed between Chinese pig breeds and European pig breeds?
Line 74, “One hundred and eighty boars“? These pigs were entire male pigs? Please provide detailed information on the age, sex, weight and genetic infor of these pigs.
How many litters do they come from?
Line 77, “Swine Feeding Standards”, What breed and age of pig does this standard apply to?
Why not use NRC 2012?
“5% or 10% GP”, Source of GP? Ingredients? Was there pre-processing?
Line 100, How old were these pigs at slaughter? Were these pigs at the normal market age for this pig breed? If they had not reached their normal market age, then the meat quality data were meaningless.
Line 163-175, 177-180, Please provide the accuracy of these ELISA kits. How many replicates were done for each sample? What about the repeatability or accuracy?
Line 183, one-way ANOVA is not the best statistic method for this study. How about the effects of pen, sex, litter or sow, and initial body weight? The author didn't even consider it?
Reviewer 2 Report
Title (please consider changing it to)
Effect of gardenia pomace supplementation on growth performance, blood metabolites, immune and antioxidant indices and meat quality in Xiangcun pigs
When you report on the effect of gardenia you should consider a logical order i.e. from the live animal to the dead (meat quality). This order should apply in the entire manuscript.
Simple summary should be written again: Do not use terms inosinic acids. Describe the main effects. Was meat quality improved? What happened with growth performance etc
Line 17: please change value to effect
Line 21: please change feeds to feed
Key words: please change to the following order
gardenia pomace; pig; growth performance; antioxidant status; immunity; meat quality.
Introduction should be written again taking into consideration the following parameters:
Description of gardenia pomace should be included. You should be clear what is meant with the term gardenia pomace. In Europe when people refer to gardenia they have in mind the flower and not the fruit.
What are the by-products of gardenia. How much it is produced. How it it preserved and what are the possible routes of utilization.
What are the characteristics of gardenia pomace i.e. bio active compounds and antinutritional factors.
Other studies with farm animals.
Please delete lines 54 to 65. The information presented here is not for the introduction but for the materials and methods where you need to add a section on gardenia pomace characteristics. I understand that this is the description of the products you used.
Materials and methods
the subchapters should be presented as follows
Animals, experimental design and diets
Growth performance
Apparent digestibility
Blood biochemistry indices - Antioxidant and immunity indices
Slaughtering and meat quality (here you should refer only to sample collection and measurements in slaughter samples). When you refer to faecal samples and blood collection you should move this information on the relevant subchapters i.e. apparent digestibility and biochemical analyses.
Meat quality: Inosinic acid determination is not related to meat quality. It is an immunity index measurement.
You should be clear how did you perform each measurement. My understanding is that you measured muscle pH and colour not in dissected samples but on the carcass. This is fine with the pH measurement but how did you measure rib eye colour in an undissected sample? You cannot do dissection at 45min after slaughter. Please provide accurate information on how you did each type of analysis.
The formula used to calculate each parameter should be placed after the presentation of each method.
Statistical analysis
You should describe the analysis clearly. For slaughter samples you had one animal from each replicate. You had six animals per treatment. So, you cannot have the mean of each replicate as an experimental unit. This refers to the feeding trial results.
Results
The results should be presented in the order described in the materials and methods section.
Table 4. Please correct to drip loss and shear force.
Line 229: please correct indexes to indices
Discussion
You have to comment on the characteristics of the gardenia pomace you used and compare it with results from other studies, where available. Then you also need to refer that there is no toxicity and this is corroborated by the finding that gardenia pomace supplementation did not have an adverse effect on growth performance and feed digestibility. You need to make this point clear.
Please try to refer to trials regarding farm animals.
Line 302: We tentatively believe that Xiangcun pigs could adapt to GP feeds.
Line 327: We speculated that this...
You do not do science with beliefs and speculations.
Lines 345 - 365: Please explain your own results.
Do your pH values show that conversion of muscle into meat was not normal? Were the animals stress susceptible? Do you have PSE or DFD meat? Colour measurement is not related to muscle mass but is related to the lower drip loss.
Conclusions
Please write again taking into consideration the aim of the study.
The colour measurements were taken at 45 min and 24h after slaughter. It is very early to discuss about oxymyoglobin and metmyoglobin as this is related to packaging, blooming etc and you do not have any data on that.
Did you do a fatty acid analysis? How can you report on n-3 PUFA.
General comment
As it stands it is very difficult to correct the manuscript. The manuscript should revised according to the comments of the reviewers in order to finalise the review.
Round 2
Reviewer 1 Report
The problems I raised earlier are basically solved after a revision by the authors. Except there are a few minor problems to note:
Line 122, “male pig”? These 180 finishing pigs were not castrated? Please double check.
Line 427, “dark, firm, and dry”
Reviewer 2 Report
The manuscript has significantly improved. However, due to the fact that there many changes it is very hard to read it. Therefore, not all changes should have been presented with the "track changes" feature.
However, there are certain points that need corrections.
There is a repetition between text present in lines 59-66 and 68 onwards. Please make your introduction focused.
-------------------------------------------------------------------------------------
Meat quality: Inosinic acid determination is not related to meat quality. It is an immunity index measurement.
Re: Thanks for your suggestion. It has been added in Slaughtering and Meat quality (See lines 302-309, 439-442 and 455-460 in the revision).
---------------------------------------------------------------------------------------
You should be clear how did you perform each measurement. My understanding is that you measured muscle pH and colour not in dissected samples but on the carcass. This is fine with the pH measurement but how did you measure rib eye colour in an undissected sample? You cannot do dissection at 45min after slaughter. Please provide accurate information on how you did each type of analysis.
Re: Thanks for your suggestion. Muscle pH and colour were measured in dissected samples. The left and right longissimus dorsi muscle tissues between the 10th and 11th ribs were taken from each slaughtered pig after 45 minutes. Accurate information is provided in Materials and Methods (See lines 310-313 in the revision).
In the correction, you are referring that colour measurement was taken after 45 min. Even after 45 min the muscle has not been converted into meat so that you can dissect the muscle. Ideally you should have measured the pH at 45 min and 24h after slaughter to check whether has the pH fall was normal and meat colour at 24 h the earliest.
It is another thing performing the pH measurement at 45 minutes with the probe pH meter and another thing doing the colour measurement that you need a flat cut surface of the rib-eye to do the measurement.
--------------------------------------------------------------------------------------
In the discussion section you are using past tense when it is not necessary. You should examine your text very carefully in order to see in which sentences past tense should be used.
For example
a) Gardenia was widely cultivated for ornamental use; moreover, it was used in traditional Chinese medicine [28,29,35,36].
Gardenia is still cultivated...
b) In general, meat with drip loss > 6.0% and pH < 5.9 601 after 45 min of slaughter was called PSE meat(pale, soft, exudative). Meat with drip loss < 602 2.0% and pH > 6.5 after 24 h of slaughter was called DFD meat(dark, firmand, dry) [54].
Meat with drip loss > 6.0% and pH < 5.9 601 after 45 min of slaughter is called and pH > 6.5 after 24 h of slaughter is called DFD meat(dark, firmand, dry) [54].
In meat science, it is still the same regarding DFD and PSE meat.
